# AdaST: Adaptive Semantic Transformation of Visual Representation for Training-free Zero-shot Composed Image Retrieval

## Abstract

Composed Image Retrieval (CIR) retrieves a target image given a reference image and a textual modification. The instruction specifies the intended change, while other visual attributes are preserved for consistency. Recent work has explored training-free methods that synthesize proxy images by combining a reference image with a textual modification. However, such methods are computationally expensive and time-consuming, while relying solely on text queries often results in the loss of crucial visual details. To address these issues, we propose Adaptive Semantic Transformation (AdaST), a new training-free method that transforms reference image features into proxy features guided by text. Instead of generating images, AdaST efficiently preserves visual information through feature-level transformation. To achieve finer-grained transformation, we introduce an adaptive weighting mechanism that balances proxy and text features, enabling the model to exploit proxy information only when it is reliable. Our method is lightweight and can be seamlessly applied to existing training-free baselines in a plug-and-play manner. Extensive experiments demonstrate that it achieves state-of-the-art performance on three CIR benchmarks while avoiding the heavy cost of image generation and incurring only marginal inference overhead compared to text-based baselines.

## 1 Introduction

Composed Image Retrieval (CIR) is the task of retrieving a target image given a reference image and a textual modification instruction. (Vo et al., 2019) The central challenge lies in multi-modal understanding and compositional reasoning, as the system must accurately integrate textual cues that specify the desired modifications with visual cues that preserve the unchanged details of the reference image. CIR has recently gained significant attention because it offers a natural and intuitive interface for exploring large-scale image collections, going beyond traditional keyword-based search. This capability is particularly important in domains such as fashion e-commerce and online search engines (Wu et al., 2021; Tian et al., 2023), where users may provide a reference product image and request modifications such as "the same shirt but in red." More broadly, CIR represents a fundamental step toward advancing vision-language understanding, as it requires aligning heterogeneous modalities and performing fine-grained reasoning that bridges visual and textual information.

Despite strong practical demand, many proprietary datasets are often kept internal and not shared with external developers. (Zhang et al., 2024; Kolouju et al., 2025) Collecting labeled CIR data, consisting of reference and target images with a modification instruction, is also costly and labor-intensive (Liu et al., 2021; Wu et al., 2021). This limits scalability and generalization to unseen domain, motivating zero-shot CIR (ZS-CIR) methods based on pretrained models without annotations. Early ZS-CIR works (Karthik et al., 2023; Yang et al., 2024b;a; Saito et al., 2023; Baldrati et al., 2023; Gu et al., 2024) treat CIR as a text-to-image retrieval task by encoding the reference image and modification instruction into a single textual representation. While effective, this strategy discards fine-grained cues, often yielding semantically correct but visually mismatched results in fashion. More recent works (Li et al., 2025; Zhou et al., 2024) synthesize a modified image via conditional generation and uses it as a retrieval input. While this preserves visual detail, genera-

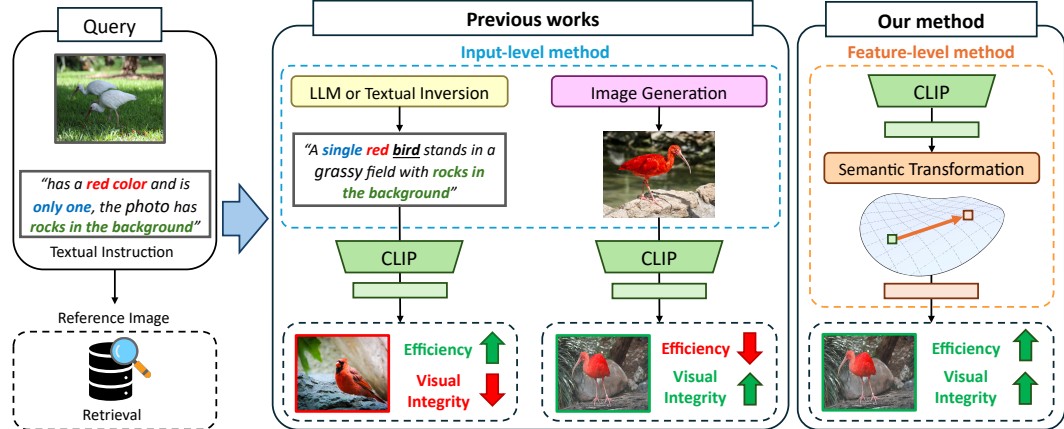

Figure 1: Comparison of existing input-level approaches and AdaST. Given a reference image and a textual instruction (*left*), prior approaches encode modifications at the input level using text or image generation (*middle*), which often sacrifice either visual integrity or efficiency. In contrast, AdaST applies instruction-guided transformations directly in the feature space of pretrained VLMs (*right*), preserving visual details while remaining efficient.

tion in high-dimensional pixel space is costly, often exceeding 30 seconds per image and surpassing usability thresholds for interactive retrieval (Nielsen, 1994).

Motivated by these limitations, we introduce **Ada**ptive **S**emantic **T**ransformation (**AdaST**), a training-free ZS-CIR method that preserves visual detail while remaining efficient and independent of external generative models. Instead of operating in pixel space or forcing the image into text space, AdaST performs instruction-guided transformations directly in the feature space of a pre-trained vision–language model (VLM) (Radford et al., 2021), as shown in Fig. 1, inspired by feature-level editing strategies (Kwon & Ye, 2022; Ye-Bin et al., 2023). Guided by LLM-generated captions, our method derives semantic shifts from text and transfers them to image embeddings, producing proxy features that better approximate the target. To ensure robustness, we further introduce an adaptive similarity mechanism that balances the contributions of proxy and text-based features, allowing the model to preserve fine-grained cues without resorting to computationally expensive image generation.

Extensive experiments on three CIR benchmarks demonstrate that AdaST not only achieves state-of-the-art performance but also remains highly efficient. In particular, AdaST improves performance by +3.47 mAP@5 over the baseline with ViT-G on the CIRCO dataset. Moreover, it runs 186× faster than state-of-the-art method IP-CIR (Li et al., 2025), while still achieving superior accuracy. Finally, our approach can be seamlessly applied to various models in a plug-and-play manner, consistently boosting their performance.

Our contributions are summarized as follows:

- We propose AdaST, a training-free ZS-CIR method that transforms reference image embeddings into proxy features guided by LLM-derived text shifts, thereby preserving fine-grained visual details without relying on generative models.

- We introduce an adaptive similarity mechanism that dynamically balances proxy- and text-based similarities, enabling the model to exploit proxy features when reliable while ensuring robustness through textual alignment.

- Extensive experiments demonstrate that AdaST achieves state-of-the-art performance on multiple CIR benchmarks while remaining substantially faster and more lightweight than generation-based methods.

## 2 RELATED WORKS

### 2.1 COMPOSED IMAGE RETRIEVAL

Composed Image Retrieval (CIR) is the task of retrieving a target image given a reference image and a textual instruction (Vo et al., 2019). This task is particularly relevant in real-world applications such as fashion e-commerce and online search engines. Prior research has focused on designing models that align image–text pairs within a shared embedding space through contrastive learning (Vo et al., 2019; Chen & Bazzani, 2020; Lee et al., 2021), or employed cross-modal attention mechanisms to capture compositional relationships (Delmas et al., 2022). However, these works rely on supervised learning using task-specific datasets (Wu et al., 2021; Liu et al., 2021; Baldrati et al., 2023), whose large-scale construction is costly and limits scalability and generalization.

### 2.2 ZERO-SHOT COMPOSED IMAGE RETRIEVAL

ZS-CIR approaches aim to reduce the cost of dataset construction by enabling of unseen images without training on CIR-specific datasets, through the utilization of large-scale pretrained VLMs (Radford et al., 2021). Early approaches encode both the reference image and the modification text into a single textual representation, typically through either textual inversion or LLM-based methods. Textual inversion methods (Saito et al., 2023; Baldrati et al., 2023; Gu et al., 2024) train an inversion model (Kumari et al., 2023; Gal et al., 2023; Ruiz et al., 2023) to map images into the text token space, which is then combined with modification instructions for retrieval. LLM-based approaches (Karthik et al., 2023; Yang et al., 2024b;a; Tang et al., 2025) instead caption images into natural language and let Large Language Models (LLMs) (Brown et al., 2020; Touvron et al., 2023) reason jointly over the reference description and the modification text. However, both approaches inevitably compress visual information into textual form, discarding fine-grained details from the reference image. Recent work (Li et al., 2025) directly synthesizes modified image from reference image and textual instruction using conditional generative models (Zhou et al., 2024; Wei et al., 2023) in pixel space. While this preserves visual detail, generation in high-dimensional pixel space is costly, due to reliance on large-scale generative models. Motivated by this limitation, we propose an efficient alternative that preserves visual fidelity without resorting to expensive image generation.

### 2.3 TEXT-GUIDED SEMANTIC TRANSFORMATION

Text-guided semantic transform methods (Fu et al., 2022; Kwon & Ye, 2022; Gal et al., 2022; Ye-Bin et al., 2023; Park et al., 2025) exploit the latent space of pretrained vision–language models (VLMs) such as CLIP to align image features with textual guidance. These methods assume that images and texts are embedded in a joint feature space where their representations are semantically aligned. In this space, the difference vector between source and target text features can be applied to the corresponding image features, yielding transformed image representations. Several CIR methods (Vo et al., 2019; Li et al., 2025) have also adopted this idea by directly applying the text-feature difference vector to image features, but such direct transfer has shown limited effectiveness. To address this limitation, we propose a novel rescaling strategy that preserves the direction of the text-feature difference vector while adapting its magnitude to the image feature space. This yields more faithful transformed image features and enables composed image retrieval that is both effective and efficient.

## 3 METHOD

We present **AdaST** (**Ada**ptive **S**emantic **T**ransformation), a training-free method for zero-shot composed image retrieval. Given a reference image $I_r$, a textual instruction $T_{\text{inst}}$, and an image database $\mathcal{D} = \{I_i^{\text{DB}}\}_{i=1}^{N}$, the goal is to retrieve the target image $I_t \in \mathcal{D}$ that reflects the modification specified by the instruction. Our approach consists of three main components. First, we generate text guidance in the form of a reference caption and a target caption using a captioning model and an LLM in section 3.1. Second, we introduce a text-guided semantic transformation that projects the semantic shift between captions into the image embedding space to construct a proxy embedding in section 3.2. Finally, we design an adaptive similarity fusion module with a gating mechanism that selectively incorporates proxy-based similarity, thereby balancing semantic alignment with visual cues for robust retrieval in section 3.3.

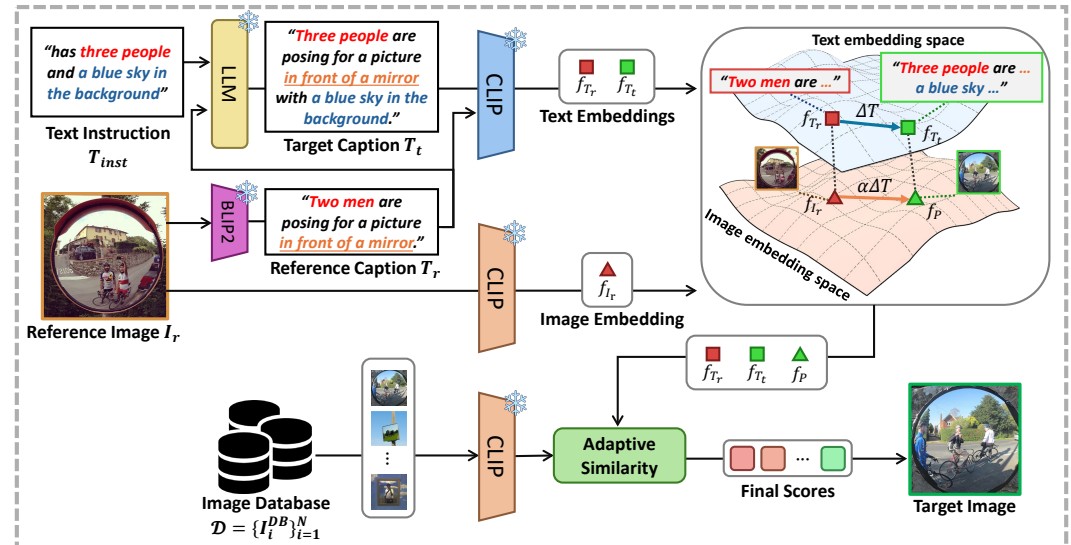

Figure 2: Overall pipeline of AdaST. It consists of three stages. (1) Text guidance generation: a reference caption is obtained from the input image using a captioning model, and an LLM combines it with the textual instruction to generate a target caption. (2) Text-guided semantic transformation: both captions and the reference image are embedded with CLIP, where the feature difference between the reference and target captions is transferred to the reference image feature with a scaling factor, yielding a proxy feature. (3) Adaptive similarity fusion: an adaptive gating mechanism fuses proxy similarity with text-based similarity, allowing proxy similarity to contribute only when supported by consistent semantic cues.

### 3.1 TEXT GUIDANCE GENERATION

We begin by constructing text guidance, which serves as a semantic bridge between the visual reference and the textual instruction, thereby facilitating accurate and robust retrieval of the modified target image. The guidance comprises two components: a reference caption $T_r$, which describes the reference image, and a target caption $T_t$, which encodes the intended modification conditioned on both $T_r$ and $T_{\text{inst}}$. The reference caption $T_r$ is obtained by passing the reference image $I_r$ through a recent captioning model such as BLIP-2 (Li et al., 2023). We then generate a target caption $T_t$ conditioned on both $T_r$ and $T_{\text{inst}}$ using recent LLM-based approaches (Karthik et al., 2023). The target caption is essential since it explicitly encodes the semantic shift specified by the instruction, ensuring that retrieval emphasizes the intended modification rather than mere visual similarity to the reference image.

### 3.2 TEXT-GUIDED SEMANTIC TRANSFORMATION

We propose a text-guided semantic transformation that transfers the semantic shift captured in the text space into the image space to construct a proxy embedding $f_P$. This method is training-free and operates directly in the feature space, enabling efficient retrieval without costly image generation while still preserving visual information. Specifically, we embed both captions and images into a joint representation space using a text encoder $E_T$ and an image encoder $E_I$ from a pretrained VLM such as CLIP (Radford et al., 2021). This yields the following embeddings:

$$f_{T_r} = E_T(T_r) , \ \ f_{T_t} = E_T(T_t) , \ \ f_{I_r} = E_I(I_r) . \tag{1}$$

Formally, the semantic shift between the target and reference caption is defined as

$$\Delta T = f_{T_t} - f_{T_r} . \tag{2}$$

The corresponding proxy embedding is defined as

$$f_P^{(\alpha)} = f_{I_r} + \alpha \, \Delta T , \tag{3}$$

where $\alpha$ is a scaling factor that controls the strength of the semantic transformation.

**Optimal Scaling**  A naive choice of the scaling factor such as $\alpha = 1$ often leads to suboptimal behavior: the proxy embedding too close to the reference image embedding and thus fails to sufficiently capture the intended modification as shown in section 4.2. To address this, we propose an optimization-based approach to obtain the optimal value of $\alpha$. The key principle is that the proxy embedding $f_P^{(\alpha)}$ should be well aligned with the target caption embedding $f_{T_t}$, ensuring that the intended modification is accurately represented. At the same time, it should remain sufficiently distinct from the reference embedding $f_{I_r}$ to prevent the retrieval from collapsing back to the original visual content. To encode these requirements, we formulate the optimization problem:

$$\alpha^* = \arg\min_{\alpha} \left( 1 - \mathrm{sim}(f_P^{(\alpha)}, f_{T_t}) + \beta \cdot \mathrm{sim}(f_P^{(\alpha)}, f_{I_r}) \right) , \tag{4}$$

where $\beta$ is a weighting coefficient that controls the influence of penalty term and $\mathrm{sim}(\cdot, \cdot)$ denotes cosine similarity. The first term encourages alignment with the target caption, while the second penalizes excessive similarity to the reference image. This objective admits a closed-form solution that can be computed directly from inner products:

$$\alpha^* = \frac{x^\top y \cdot x^\top d - d^\top y \cdot \|x\|^2}{d^\top y \cdot x^\top d - x^\top y \cdot \|d\|^2}, \tag{5}$$

where $x = f_{I_r}$, $y = \tilde{f}_{T_t} - \beta \tilde{f}_{I_r}$, and $d = \Delta T$. Here, $\tilde{f} = f/\|f\|_2$. The resulting proxy embedding $f_P^{(\alpha^*)}$ effectively captures the semantic shift specified by the textual instruction while remaining anchored in the visual space of the reference image.

### 3.3  ADAPTIVE SIMILARITY FUSION

While the proxy embedding can be directly used for retrieval by measuring its similarity to candidate image features, relying solely on proxy-based similarity introduces a drawback. Since it primarily captures visual cues, the proxy may assign high scores to images that are visually similar yet semantically irrelevant to the instruction. To address this issue, we incorporate semantic guidance through the target caption feature $f_{T_t}$, which encodes rich semantic information. We compute target-caption similarity between target caption feature and the candidate features, and then fuse this score with the proxy-based similarity.

Specifically, we propose a gating mechanism inspired by (Yang et al., 2024b;a) that adaptively regulates the contribution of proxy similarity. This gating strategy ensures that proxy-based similarity influences retrieval only when supported by semantic evidence, thereby reducing false positives caused by purely visual resemblance. To implement this, we first extract feature of the database images:

$$f_{I_i^{\mathrm{DB}}} = E_I(I_i^{\mathrm{DB}}), \ \ \forall i = \{1, \ldots, N\} . \tag{6}$$

For simplicity, let $f_{I^{\mathrm{DB}}}$ denote the set of all database embeddings. For each query, we then compute three similarity scores:

$$S_{T_t} = \mathrm{sim}(f_{T_t}, f_{I^{\mathrm{DB}}}) , \ \ S_{T_r} = \mathrm{sim}(f_{T_r}, f_{I^{\mathrm{DB}}}) , \ \ S_P = \mathrm{sim}(f_P, f_{I^{\mathrm{DB}}}) . \tag{7}$$

The proposed gating function is defined as

$$G(\Delta S_T) = \begin{cases} \lambda, & \Delta S_T + m \geq 0 \\ 0 , & \text{otherwise} \end{cases} , \ \ \ \Delta S_T = S_{T_t} - S_{T_r} , \tag{8}$$

where $\lambda$ is a weighting coefficient that controls the influence of $S_P$ and $m$ is a margin that determines whether semantic alignment is achieved. The gate activates only when the target caption shows greater semantic alignment than the reference, ensuring that proxy-based similarity is incorporated only under supportive evidence.

Under this regulation, the final similarity score is given by

$$S_{\text{total}} = S_A \cdot S_P + S_{T_t} , \ \ \ S_A = S_{T_t} \cdot G(\Delta S_T) , \tag{9}$$

where $S_A$ represents an adaptive weight on the proxy similarity $S_P$. Finally, the retrieval result is obtained by selecting the database image with the maximum similarity score:

$$I_t = \arg\max_{I_i^{\mathrm{DB}} \in \mathcal{D}} S_{\text{total}} . \tag{10}$$

Table 1: Quantitative results on the CIRCO and CIRR benchmarks using three backbones (ViT-B/32, ViT-L/14, and ViT-G/14), where our method is applied on top of two representative baselines (CIReVL (Karthik et al., 2023) and SEIZE (Yang et al., 2024a)). Across all three architectures and both benchmarks, combining our approach with the baselines yields consistent improvements and achieves state-of-the-art performance on most metrics. † represents our reproduced results.

| Benchmark | | CIRCO (mAP@K) | | | | CIRR (Recall@K) | | | | CIRR (Recall_subset@K) | | |
|---|---|---|---|---|---|---|---|---|---|---|---|---|
| Backbone | Method | k=5 | k=10 | k=25 | k=50 | k=1 | k=5 | k=10 | k=50 | k=1 | k=2 | k=3 |
| ViT-B/32 | SEARLE ICCV23 | 9.35 | 9.94 | 11.13 | 11.84 | 24 | 53.42 | 66.82 | 59.78 | 54.89 | 76.60 | 88.19 |
| | CIReVL† ICLR24 | 13.28 | 13.69 | 15.13 | 15.94 | 20.84 | 46.96 | 60.19 | 84.70 | 54.30 | 76.5 | 88.10 |
| | LDRE SIGIR24 | 17.96 | 18.32 | 20.21 | 21.11 | 25.69 | 55.13 | 69.04 | 89.9 | 60.53 | 80.65 | 90.7 |
| | SEIZE† ACMMM24 | 18.75 | 19.37 | 21.09 | 22.07 | 26.96 | 55.59 | 68.24 | 88.34 | 66.82 | 85.23 | 93.35 |
| | OSrCIR CVPR25 | 18.04 | 19.17 | 20.94 | 21.85 | 25.42 | 54.54 | 68.19 | - | 62.31 | 80.86 | 91.13 |
| | CIReVL† + Ours | 15.20 | 15.73 | 17.25 | 18.12 | 25.23 | 52.41 | 64.48 | 85.35 | 60.12 | 78.96 | 89.11 |
| | SEIZE† + Ours | 21.16 | 21.89 | 23.76 | 24.62 | 30.15 | 59.71 | 72.60 | 89.81 | 66.72 | 84.94 | 93.45 |
| ViT-L/14 | Pic2Word CVPR23 | 8.72 | 9.51 | 10.64 | 11.29 | 23.9 | 51.7 | 65.3 | 87.8 | - | - | - |
| | SEARLE ICCV23 | 11.68 | 12.73 | 14.33 | 15.12 | 24.24 | 52.48 | 66.29 | 88.84 | 53.76 | 75.01 | 88.19 |
| | LinCIR CVPR24 | 12.59 | 13.58 | 15.00 | 15.85 | 25.04 | 53.25 | 66.68 | - | 57.11 | 77.37 | 88.89 |
| | CIReVL† ICLR24 | 16.54 | 17.42 | 19.27 | 20.22 | 21.28 | 47.47 | 60.6 | 83.4 | 54.5 | 75.28 | 87.88 |
| | LDRE SIGIR24 | 23.35 | 24.03 | 26.44 | 27.5 | 26.53 | 55.57 | 67.54 | 88.5 | 60.43 | 80.31 | 89.9 |
| | SEIZE† ACMMM24 | 24.71 | 25.52 | 27.99 | 29.03 | 28.43 | 56.53 | 69.88 | 88.17 | 66.43 | 84.68 | 92.96 |
| | OSrCIR CVPR25 | 23.87 | 25.33 | 27.84 | 28.97 | 29.45 | 57.68 | 69.86 | - | 62.12 | 81.92 | 91.10 |
| | LDRE + IP-CIR CVPR25 | 26.43 | 27.41 | 29.87 | 31.07 | 29.76 | 58.82 | 71.21 | 90.41 | 62.48 | 81.64 | 90.89 |
| | CIReVL† + Ours | 20.32 | 20.92 | 22.81 | 23.71 | 25.35 | 52.92 | 66.41 | 86.89 | 60.75 | 80.77 | 90.92 |
| | SEIZE† + Ours | 28.94 | 29.65 | 32.04 | 33.03 | 30.72 | 59.78 | 71.13 | 88.68 | 67.21 | 84.96 | 93.04 |
| ViT-G/14 | LinCIR CVPR24 | 19.71 | 21.01 | 23.13 | 24.18 | 35.25 | 64.72 | 76.05 | - | 63.35 | 82.22 | 91.98 |
| | CIReVL† ICLR24 | 26.47 | 27.46 | 29.91 | 30.86 | 30.7 | 59.66 | 70.89 | 89.86 | 63.54 | 82.02 | 91.52 |
| | LDRE SIGIR24 | 31.12 | 32.24 | 34.95 | 36.03 | 36.15 | 66.39 | 77.25 | 93.95 | 68.82 | 85.66 | 93.76 |
| | SEIZE† ACMMM24 | 35.61 | 36.92 | 39.67 | 40.61 | 40.87 | 69.52 | 78.94 | 92.27 | 75.04 | 90.31 | 96.02 |
| | OSrCIR CVPR25 | 30.47 | 31.14 | 35.03 | 36.59 | 37.26 | 67.25 | 77.33 | - | 69.22 | 85.28 | 93.55 |
| | LDRE + IP-CIR CVPR25 | 32.75 | 34.26 | 36.86 | 38.03 | 39.25 | 70.07 | 80.00 | 94.89 | 69.95 | 86.87 | 94.22 |
| | CIReVL† + Ours | 32.32 | 33.49 | 35.98 | 36.81 | 35.04 | 65.06 | 75.98 | 91.57 | 65.57 | 83.52 | 92.53 |
| | SEIZE† + Ours | 39.08 | 39.93 | 42.53 | 43.34 | 42.84 | 72.29 | 80.82 | 93.28 | 74.82 | 89.95 | 96.00 |

## 4 EXPERIMENTS

**CIRR** (Liu et al., 2021) consists of 21,552 images collected from the NLVR dataset (Suhr et al., 2018) and along with 36,554 associated queries. It is designed to support fine-grained natural language modifications, enabling retrieval based on subtle semantic differences between images. A limitation of CIRR is the presence of potential false negatives, as multiple images in the gallery may satisfy the same instruction, yet only one is annotated as the ground truth. **CIRCO** (Baldrati et al., 2023) is a benchmarking dataset constructed from COCO 2017 (Lin et al., 2014), explicitly addressing this false-negative issue by providing multiple annotated target images per query. It includes a validation set with 220 queries and a test set with 800 queries, covering instructions that involve attribute edits, object substitutions, and style modifications, which are particularly challenging for compositional reasoning. **Fashion-IQ** (Wu et al., 2021) is a domain-specific benchmark for fashion retrieval, containing 30,135 queries and 77,683 product images across three categories: *Shirt*, *Dress*, and *Toptee*. Its queries are written by annotators and describe modifications to reference garments.

For evaluation, we follow the official protocol from each dataset. CIRR is evaluated using Recall@K ($K \in \{1, 5, 10, 50\}$) and RecallSubset@K ($K \in \{1, 2, 3\}$), where the latter focuses only on images within the same semantic set as the reference and target, thus capturing retrieval performance under closely related distractor groups. CIRCO is evaluated with mean Average Precision at top-K results (mAP@K, $K \in \{5, 10, 25, 50\}$) due to its multiple positive targets. Fashion-IQ is evaluated with Recall@10 and Recall@50 for each garment category and their average.

**Implementation Details.** For the retrieval model, we adopt CLIP backbones including ViT-B/32, ViT-L/14, and ViT-G/14. We use the official OpenAI (Radford et al., 2021) weights by default, while for ViT-G/14 we use OpenCLIP (Ilharco et al., 2021) weights. We use CIReVL (Karthik et al., 2023) and SEIZE (Yang et al., 2024a) as baseline models for comparison. For the Fashion-IQ dataset, we additionally employ LinCIR (Gu et al., 2024) as our baseline, and additionally apply an LLM-based caption generation process. For image captioning, we employ the BLIP-2 (Li et al., 2023) model. Regarding the LLM model, the baseline codebases use GPT-3.5-turbo, which is no longer available. Therefore, following a recent work (Tang et al., 2025), we re-implemented the baseline with GPT-4o, ensuring that both our method and the baselines are evaluated under the same model for a fair comparison. All experiments are conducted on a single A6000 GPU. For all baselines and datasets, we set $\beta = 0.25$, $\lambda = 4$, and $m = 0.1$, except that $m = 0$ is used for the CIRCO dataset.

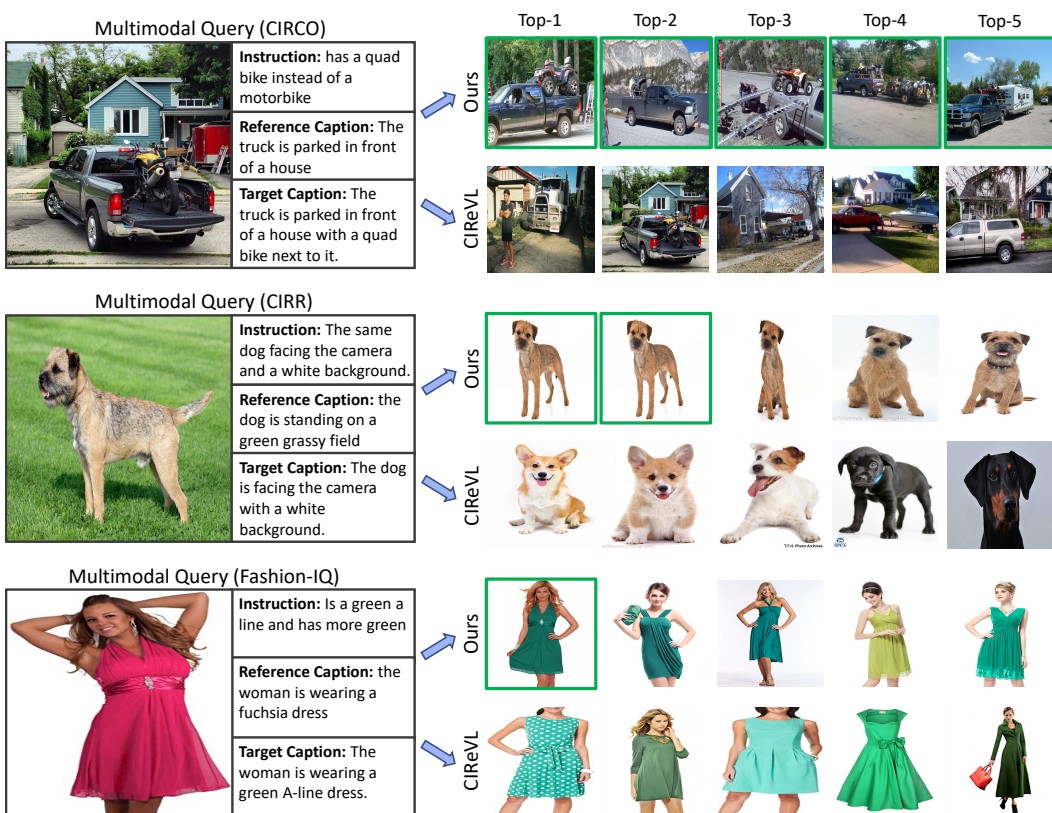

Figure 3: Qualitative comparison between CIReVL and our method on three benchmarks (CIRCO, CIRR, and Fashion-IQ). Given a reference image and an instruction, reference and target captions are generated, and the top-5 retrieved images from each method are shown, with ground-truth targets highlighted in green. Our method leverages visual features more effectively, enabling accurate retrieval even when the target caption is underspecified, by exploiting fine-grained details such as dog breeds or dress shapes.

## 4.1 COMPARISON WITH STATE-OF-THE-ART METHODS

**CIRCO and CIRR**   As shown in Tab. 1, our method consistently improves over both CIReVL and SEIZE baselines across all backbones, achieving state-of-the-art performance on most metrics. The improvements are particularly significant with large backbones such as ViT-G/14, showing the scalability of AdaST. On the CIRCO benchmark, our approach yeilds substantial improvement. For example, with ViT-G/14 backbone, our method surpasses the CIReVL baseline by +5.85 mAP@5, which is substantially larger than the gain of +1.63 mAP@5 achieved by LDRE + IP-CIR. Similarly, when applied to SEIZE, our method achieves a further gain of +3.47 mAP@5, demonstrating its effectiveness on multi-target scenario. This highlights the effectiveness of the proposed feature-level transformation in exploiting visual cues from the reference image.

Fig. 3 presents a qualitative comparison between our method and CIReVL. In the CIRCO example, our model fuses cues from the reference image with the target caption and retrieves images where a quad bike appears on (or next to) the truck at the top ranks. In contrast, CIReVL, which uses only the target caption for retrieval, often misses one of the two key objects or even returns the reference image itself. In the CIRR example, the target caption omits the breed, so CIReVL retrieves dogs that are not the "same dog," whereas our method leverages visual evidence to return the same breed as the reference (Border Terrier); note that Top-1 and Top-2 are duplicates due to dataset noise.

**Fashion-IQ**   As shown in Tab. 2, across all three architectures and both benchmarks, combining our approach with the baselines yields consistent improvements and achieves state-of-the-art performance on most metrics. Note that LinCIR is a training-based baseline leveraging textual inversion,

Table 2: Quantitative results on the Fashion-IQ benchmark using the ViT-G/14 backbone, where our method is applied on top of three representative baselines (CIReVL (Karthik et al., 2023), SEIZE (Yang et al., 2024a), and LinCIR (Gu et al., 2024)). Across all three categories (*Shirt*, *Dress*, and *Toptee*), our method combined with the baselines shows consistent improvements and reaches state-of-the-art performance on the majority of metrics. † indicates our reproduced results.

| Type | | Shirt | | Dress | | Toptee | | Average | |
|---|---|---|---|---|---|---|---|---|---|
| Backbone | Method | R@10 | R@50 | R@10 | R@50 | R@10 | R@50 | R@10 | R@50 |
| ViT-G/14 | Pic2Word    CVPR23 | 33.17 | 50.39 | 25.43 | 47.65 | 35.24 | 57.62 | 31.28 | 51.89 |
| | SEARLE    ICCV23 | 36.46 | 55.35 | 28.16 | 50.32 | 39.83 | 61.45 | 34.81 | 55.71 |
| | LinCIR    CVPR24 | 46.76 | 65.11 | 38.08 | 60.88 | _50.48_ | 71.09 | 45.11 | 65.69 |
| | CIReVL†    ICLR24 | 35.13 | 52.65 | 27.52 | 49.03 | 37.33 | 58.75 | 33.33 | 53.48 |
| | LDRE    SIGIR24 | 35.94 | 58.58 | 26.11 | 51.12 | 35.42 | 56.67 | 32.49 | 55.46 |
| | SEIZE†    ACMMM24 | 39.50 | 57.65 | 33.37 | 55.88 | 41.66 | 64.20 | 38.12 | 59.24 |
| | OSrCIR    CVPR25 | 38.65 | 54.71 | 33.02 | 54.78 | 41.04 | 61.83 | 37.57 | 57.11 |
| | LinCIR + IP-CIR    CVPR25 | _48.04_ | _66.68_ | 39.02 | 61.03 | 50.18 | _71.14_ | _45.74_ | _66.28_ |
| | CIReVL†+Ours | 40.38 | 59.08 | 36.49 | 58.70 | 43.65 | 64.10 | 40.17 | 60.63 |
| | SEIZE†+Ours | 44.36 | 62.22 | _40.21_ | _62.12_ | 48.55 | 69.30 | 44.37 | 64.55 |
| | LinCIR+Ours | **48.28** | **67.17** | **43.53** | **64.70** | **52.68** | **73.13** | **48.16** | **68.33** |

Table 3: Comparison of inference time.

| Dataset | Fashion-IQ Dress (DB size = 4K) | | CIRCO (DB size = 120K) | |
|---|---|---|---|---|
| | time | $+\Delta t$ | time | $+\Delta t$ |
| CIReVL | 1.76s | – | 2.16s | – |
| +Ours | 1.87s | 0.11s | 2.77s | 0.61s |
| +IP-CIR | 119.82s | 118.06s | 120.84s | 118.68s |
| SEIZE | 26.05s | – | 26.25s | – |
| +Ours | 26.17s | 0.12s | 26.89s | 0.64s |
| +IP-CIR | 144.19s | 118.14s | 145.21s | 118.96s |

Table 4: Ablation study on CIRCO dataset.

| Method | Proxy | Scaling | Gating | mAP@5 | Δ |
|---|---|---|---|---|---|
| CIReVL | | | | 26.47 | - |
| | ✓ | | | 27.42 | +0.95 |
| | ✓ | ✓ | | 29.41 | +2.94 |
| | ✓ | ✓ | ✓ | **32.32** | +5.85 |

which explains its relatively strong performance; when combined with our method, it achieves even higher performance. This confirms that, even in the fashion domain where natural language modifications are highly fine-grained and diverse, the proposed instruction-guided feature transformation proves effective in leveraging reference image cues. Furthermore, the qualitative results of Fashion-IQ in Fig. 3 highlight that our method captures texture and silhouette information from the reference image that is difficult to express purely in text, producing more faithful matches for instructions such as "green A-line dress."

**Inference Time** We further evaluate the efficiency of our method compared to generation-based approach (Li et al., 2025). As shown in Tab. 3, our method introduces only a negligible overhead compared to the baseline retrieval models (CIReVL and SEIZE). Specifically, on Fashion-IQ Dress (4K database), the additional computation is only 0.11–0.12 seconds, and on CIRCO (120K database), the overhead remains within 0.61–0.64 seconds. In contrast, generation-based IP-CIR requires more than 118 seconds of additional processing time on both datasets, making it over two orders of magnitude slower. These results clearly demonstrate that our approach maintains near real-time efficiency while substantially improving accuracy. By avoiding costly image generation, our method scales effectively to large databases and provides a practical alternative to generation-based solutions.

## 4.2 ABLATION STUDY

**Component Analysis** To gain deeper insight into the impact of each component, we conduct an ablation study to evaluate the effectiveness of each module in our framework: proxy embedding, optimal scaling, and the gating function, as shown in Tab. 4. The baseline model (CIReVL) shows 26.47 mAP@5. Using proxy embedding alone ($\alpha = 1$) achieves 27.42 mAP@5 (+0.95). Incorporating the scaling strategy further improves performance to 29.41 mAP@5 (+2.94). Finally, introducing the gating function in conjunction with proxy and scaling yields the best result of 32.32 mAP@5 (+5.85). These results confirm that adaptively regulating the contribution of proxy similarity is crucial for suppressing misleading visual cues and enhancing retrieval accuracy.

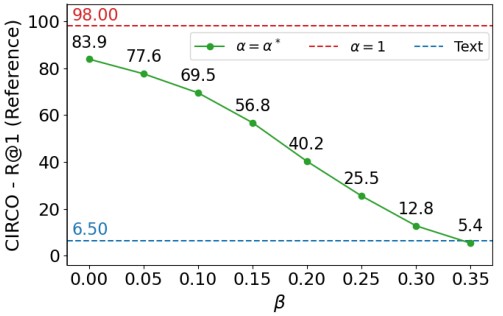
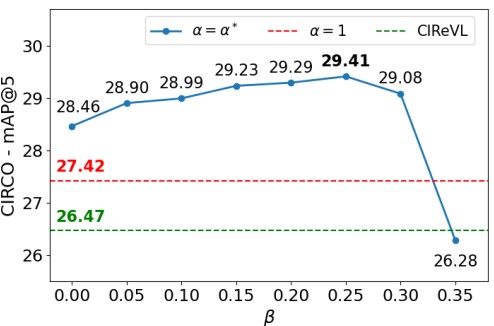

Figure 4: Retrieval performance with the reference image as ground truth.

Figure 5: Ablation study of optimal scaling.

**Analysis for the Optimal Scaling of Semantic Transformation**   To better understand the behavior of the proxy embedding, we conduct a controlled retrieval experiment on the CIRCO dataset where the objective is to retrieve the reference image rather than the target. This experiment enables us to directly examine how much the proxy embedding diverges from the original reference representation. As shown in Fig. 4, the proxy mostly fails to sufficiently deviate from the reference. In particular, when the semantic shift from the text space is naively transferred, the resulting proxy embedding achieves an R@1 of 98.0, indicating that it remains largely unchanged from the reference. This finding motivates us to explore an optimal scaling strategy.

According to the results, we find that enforcing only the first condition, which requires the proxy embedding to align with the target caption embedding (controlled by $\alpha$), is insufficient. Although this condition pushes the proxy embedding away from the reference, it still remains in its vicinity. By introducing an additional parameter $\beta$, we observe that the proxy embedding gradually diverges further from the reference image, and this divergence is closely tied to retrieval performance, as shown in Fig. 5. When $\alpha$ is fixed to 1, the performance improves over the baseline but does not yield a substantial gain. In contrast, applying our proposed scaling strategy leads to a consistent increase in performance until $\beta$ exceeds a certain threshold, at which point performance drops sharply. This degradation occurs because the proxy embedding becomes overly detached from the reference, which is consistent with the intuition that the target image still preserves essential information from the reference. Furthermore, when we directly use the reference image ($\alpha = 0$), the performance drops significantly, achieving only mAP@5 of 22.75. Unlike the proxy embedding, which remains close to the reference but shifts toward the target direction, this result demonstrates that the semantic transformation is crucial for effectively guiding the proxy embedding.

## 5   CONCLUSION

In this work, we proposed Adaptive Semantic Transformation (AdaST), a training-free method for Composed Image Retrieval that achieves a new state-of-the-art in both accuracy and efficiency. We identified a core challenge in existing methods: a forced choice between the efficiency of text-based approaches, which lose visual detail, and the fidelity of generation-based methods, which are computationally expensive. AdaST resolves this dilemma by transforming reference image features directly in the latent space, guided by textual instructions. This approach effectively preserves fine-grained visual information without resorting to costly image synthesis. Our experiments on three benchmarks demonstrate that AdaST significantly outperforms previous methods. The introduction of an adaptive similarity mechanism further improves robustness by intelligently weighting visual and textual cues. As a result, AdaST is not only more accurate but also substantially faster than generation-based alternatives, making it highly suitable for practical applications. Its modular, plug-and-play design also allows for easy integration into existing ZS-CIR pipelines. We believe our feature-space transformation approach offers a promising and efficient direction for the future of multi-modal retrieval and understanding.

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

# A APPENDIX

## A.1 ETHICS STATEMENT

Following the ICLR 2026 guidelines, we disclose that a Large Language Model (LLM) was used during the preparation of this manuscript for grammar correction, text refinement, manuscript review, and related research searches through Deep Research. The LLM was also employed to support experiments and to generate code for plotting figures (Fig. 4 and Fig. 5) based on the data obtained from our experiments. All research contributions, experimental results, and scientific claims are entirely the responsibility of the authors.

