# OpenReview forum: "AdaST: Adaptive Semantic Transformation of Visual Representation for Training-free Zero-shot Composed Image Retrieval"
_ICLR.cc/2026/Conference — ICLR 2026 Conference Withdrawn Submission_

### Official Review · Reviewer_garP · 2025-10-30

**Soundness:** 3
**Presentation:** 3
**Contribution:** 3
**Rating:** 6
**Confidence:** 4

**Summary:**

This paper proposes Adaptive Semantic Transformation (AdaST), a training-free method for composed image retrieval. The core idea is to transform the reference image into a proxy feature embedding guided by the difference between the reference caption and the target caption. The similarity score based on the proxy feature is incorporated into the original scores via a gating scheme. Experiments on several datasets demonstrate the effectiveness and efficiency of the proposed method.

**Strengths:**

1. The paper is well written, with clear background, method descriptions, and experimental analysis.

2. The method is highly adaptable and can be integrated into existing approaches in a plug-and-play manner.

3. The method operates directly in feature space, runs fast, and has strong practical value.

**Weaknesses:**

1. The paper demonstrates the effectiveness of the proxy feature mainly through experimental and visualization results, but lacks deeper or alternative explanations, such as feature visualizations and statistics of score distributions before and after applying proxy similarity.

2. The ablation study is relatively simple and lacks ablations on other hyperparameters.

3. There is no analysis of failure cases: in what situations is the proxy most effective, and in what situations might it have negative effects?

**Questions:**

The designs of Equations (8) and (9) are quite intuitive. What are their design principles and specific meanings? For example, in Equation (9), why is the adaptive weight multiplied by $S_{T_t}$? Are there other options?

---

### Official Review · Reviewer_nn1V · 2025-10-31

**Soundness:** 3
**Presentation:** 3
**Contribution:** 3
**Rating:** 6
**Confidence:** 5

**Summary:**

The paper presents a training-free zero-shot combined image retrieval method called Adaptive Semantic Transformation(AdaST), aimed at addressing the trade-off between visual detail preservation and computational efficiency in existing methods. AdaST does not rely on image generation, but instead performs semantic transformations of reference image features through text guidance in the feature space of a pre-trained vision-language model, generating a "proxy feature" to approximate the target image. The method also introduces an adaptive similarity fusion mechanism that dynamically balances the contributions of proxy features and text features, achieving state-of-the-art performance across multiple CIR benchmarks while significantly improving inference speed.

**Strengths:**

1. The paper is training-free, relying entirely on a pre-trained model without the need for additional labeled data or fine-tuning.
2. Extensive experiments demonstrate efficient inference by avoiding the high-cost image generation process, significantly outperforming generative methods in terms of inference speed.
3. The method is plug-and-play, seamlessly integrating into existing training-free CIR methods, enhancing their performance.

**Weaknesses:**

1. The gating threshold m, weight λ, and β all require manual grid-search, yet the main text describes them as "adaptive," which is not fully accurate.
2. The influence of different LLMs on the results has not been adequately explored.
3. There is a spelling error in the term "yeilds" in the experimental section, which should be "yields"; also, "domain" in the INTRODUCTION may need to be changed to the plural form "domains". It is recommended to carefully review the entire manuscript for such issues.
4. The plus signs in Table 2 have inconsistent styles; it is advised to check and adjust the formatting throughout the document to ensure consistency and conformity.
5. The manuscript lacks examples of LLM prompts; it is recommended to include relevant examples to enhance reproducibility and transparency.

**Questions:**

1. I hope the authors can address each of the weaknesses and explain how this work is not merely repetitive.
2. What is the difference from SEIZE? It is necessary to clarify the similarities and differences between AdaST and SEIZE in terms of feature transformation, similarity fusion, and other aspects.
3. How does AdaST compare qualitatively with SEIZE? Could more visual examples be provided to demonstrate AdaST's advantages in detail preservation and semantic alignment?
4. Will the code be open-sourced? The release of the code is crucial for reproducibility and credibility. Is there a plan for publication?
5. Does "plug-and-play" imply post-processing? Is it subsequent processing based on the output of the original method, or does it fully replace the original process?

---

### Official Review · Reviewer_BEwH · 2025-11-01

**Soundness:** 2
**Presentation:** 2
**Contribution:** 2
**Rating:** 2
**Confidence:** 4

**Summary:**

This paper introduces AdaST, a method that predicts target-image features for two-stage training-free CIR. The authors aim to leverage a text-driven manifold augmentation strategy to avoid the high cost of pixel-level generation in ZS-CIR. Experiments on CIRCO, CIRR, and Fashion-IQ report SOTA performance while running ~186× faster than pixel-level, generation-based IP-CIR.

**Strengths:**

1.	The idea of this paper is intuitive.
2.	It is interesting to predict target-image information at the feature level rather than the pixel level.
3.	Experiments on CIRCO, CIRR, and Fashion-IQ achieve SOTA performance.

**Weaknesses:**

1.	Lack of novelty. The motivation of this paper, which predicts target information at the feature level instead of the pixel level, has been explored in PrediCIR [1], which is more efficient than the proposed method; the authors overlook a proper comparison and acknowledgement. This raises concerns that the paper does not offer sufficient new insight to the community.
2.	Limited technology contribution. While feature-level target prediction has been studied [1], this paper appears to adopt TextManiA’s [2] text-driven manifold augmentation to realize it. Moreover, the key module, the Adaptive Similarity module, closely mirrors LDRE’s [3] Adaptive Semantic Ensemble (Sec. 4.2.2). Thus, the technical contribution seems insufficient.
3.	Concerns of the generalizability. he method seems restricted to a two-stage, training-free CIR pipeline. Recent work suggests one-stage approaches are the trend in training-free CIR, offering higher efficiency and avoiding information loss [3]. This further weakens the contribution.
4.	Concerns about the hallucination issues. The output of AdaST is based on VLM (i.e.,BLIP-2), a MLLM-generated results without CoT process, which make me concern the hallucination problem. Such issues could significantly impact the retrieval results. A more detailed analysis of hallucination risks is needed.
5.	Need more analysis experiments. For example, have a visualization experiment to show the benefit of the text-driven manifold augmentation (e.g., add a decoder to generate the corresponding target image).
6.	Missing implementary details. The pipeline has many components, and the code is not provided. Including pseudocode is recommended.
7.	Insufficient ablation studies. For example, what is the performance with open-source MLLM (i.e., Qwen-VL-72B)? What is the influence of other gating methods? What is the influence of other Optimal Scaling methods?
8.	Incomplete benchmarking. Comparisons on CLIP-B/L for Fashion-IQ are missing, although these are common in recent ZS-CIR evaluations.

Overall, the novelty and technical contribution appear limited, and analyses on generalizability and hallucination are missing. Therefore, I gave the Reject recommendation. I believe the paper should have a revision to address these concerns.

References

[1] Tang Y, Yu J, Gai K, et al. Missing target-relevant information prediction with world model for accurate zero-shot composed image retrieval[C]//Proceedings of the Computer Vision and Pattern Recognition Conference. 2025: 24785-24795.

[2] Ye-Bin M, Kim J, Kim H, et al. Textmania: Enriching visual feature by text-driven manifold augmentation[C]//Proceedings of the IEEE/CVF International Conference on Computer Vision. 2023: 2526-2537.

[3] Sun Z, Jing D, Lu Z. CoTMR: Chain-of-Thought Multi-Scale Reasoning for Training-Free Zero-Shot Composed Image Retrieval[J]. arXiv preprint arXiv:2502.20826, 2025.

**Questions:**

1.	Can the proposed method be used in one-stage, training-free CIR?
2.	Why not compare against feature-level target-prediction methods for CIR?
3.	What is the performance with open-source MLLMs (e.g., Qwen-VL-72B)?
4.	What is the influence of alternative gating methods?
5.	What is the influence of different optimal-scaling strategies?
6.	What is the performance on CLIP-B/L for Fashion-IQ?

---

### Official Review · Reviewer_Hjcc · 2025-11-04

**Soundness:** 3
**Presentation:** 3
**Contribution:** 2
**Rating:** 4
**Confidence:** 4

**Summary:**

This paper proposes the training-free Adaptive Semantic Transformation (AdsST) approach for the task of composed image retrival, which transforms reference image features into proxy features with the guidance of text, preserving visual information through feature-level transformation.

**Strengths:**

1. The paper is well organized and easy to follow.
2. Extensive abaltion expereiments are conducted to demonstrate the effectiveness of the proposed AdaST method.

**Weaknesses:**

1. The paper claims that existing methods are computationally expensive, but no efficiency evaluations are presented to show AdaST's improvements on computational cost.
2. The architecture of the proposed AdaST method is not new, more theoretical insight should be provided to strength the novelty of AdaST.

**Questions:**

1. Efficiency evaluations on computational cost should be provided to validate the effectiveness of AdaST.
2. More theoretical insight of the proposed method should be presented to strength the novelty of AdaST.

---

### Note · Authors · 2025-11-13

I have read and agree with the venue's withdrawal policy on behalf of myself and my co-authors.